# Enhancing Transcriptional Reprogramming of Mesenchymal Glioblastoma with Grainyhead-like 2 and HDAC Inhibitors Leads to Apoptosis and Cell-Cycle Dysregulation

**DOI:** 10.3390/genes14091787

**Published:** 2023-09-12

**Authors:** Spandana Kotian, Rachel M. Carnes, Josh L. Stern

**Affiliations:** Department of Biochemistry and Molecular Genetics, University of Alabama at Birmingham, Birmingham, AL 35233, USA

**Keywords:** glioblastoma, GRHL2, HDAC, mesenchymal, EMT, apoptosis, cell cycle, aneuploidy

## Abstract

Glioblastoma (GBM) tumor cells exhibit mesenchymal properties which are thought to play significant roles in therapeutic resistance and tumor recurrence. An important question is whether impairment of the mesenchymal state of GBM can sensitize these tumors to therapeutic intervention. HDAC inhibitors (HDACi) are being tested in GBM for their ability promote mesenchymal-to-epithelial transcriptional (MET) reprogramming, and for their cancer-specific ability to dysregulate the cell cycle and induce apoptosis. We set out to enhance the transcriptional reprogramming and apoptotic effects of HDACi in GBM by introducing an epithelial transcription factor, Grainyhead-like 2 (GRHL2), to specifically counter the mesenchymal state. GRHL2 significantly enhanced HDACi-mediated MET reprogramming. Surprisingly, we found that inducing GRHL2 in glioma stem cells (GSCs) altered cell-cycle drivers and promoted aneuploidy. Mass spectrometry analysis of GRHL2 interacting proteins revealed association with several key mitotic factors, suggesting their exogenous expression disrupted the established mitotic program in GBM. Associated with this cell-cycle dysregulation, the combination of GRHL2 and HDACi induced elevated levels of apoptosis. The key implication of our study is that although genetic strategies to repress the mesenchymal properties of glioblastoma may be effective, biological interactions of epithelial factors in mesenchymal cancer cells may dysregulate normal homeostatic cellular mechanisms.

## 1. Introduction

Glioblastoma (GBM) is the most common malignant primary brain tumor in patients [1]. The current standard of care is surgical resection of the tumor followed by radiation and chemotherapy with the DNA alkylating agent temozolomide (TMZ) [2,3]. Even with this treatment, approximately 90% of patients undergo recurrence within five years post-diagnosis [4]. There is a pressing need to develop new therapeutic approaches for GBM.

While GBM tumors display profound intercellular heterogeneity that is dynamic, almost all GBM cells express key drivers of the mesenchymal phenotype, such as MMP2, SNAI2, YKL-40 (*CHI3L1*), and ZEB1 [5,6,7,8]. In addition, recurrent GBM tumors appear to express elevated levels of select mesenchymal factors [5,9]. Three major GBM cellular subtypes have been characterized [10,11,12], and the mesenchymal subtype commonly displays the worst prognosis [13,14,15]. In addition to therapeutic resistance, the mesenchymal properties of GBM facilitate invasion of the surrounding environment, making total surgical extirpation impossible [16,17]. These observations suggest that methods to impair the mesenchymal phenotype of GBM tumors may promote therapeutic approaches. The feasibility of manipulating the mesenchymal state for potential patient benefit is supported by many reports, including the observation that engineered loss of ZEB1 through genetic or pharmacological methods increased or created novel drug vulnerabilities [5,18,19].

Grainyhead-like 2 (GRHL2) is a developmental transcription factor associated with the epithelial phenotype [20,21]. In some cells, it is a direct positive regulator of E-cadherin (*CDH1*) expression [21]. As such, in normal epithelial cells, GRHL2 plays a role in establishing apical–basal polarity [22], which relies on E-cadherin and adherens junctions. GRHL2 plays a critical role in oriented cell division and the positioning of the mitotic spindle [23], and its expression in epithelial cells likely represents a barrier to epithelial-to-mesenchymal transition in cancer [24,25]. Indeed, GRHL2 inhibits the epithelial-to-mesenchymal transition (EMT) in some cancer cells [26,27,28]. However, the role of GRHL2 in oncogenesis varies in different cancer types. In bladder and gastric cancers, GRHL2 acts as a tumor suppressor by directly repressing *ZEB1* mRNA expression [20,27], while in breast, ovarian, and colorectal cancers, GRHL2 has been shown to induce resistance to apoptosis by downregulating the death receptor FAS [29], demonstrating tumor-specific roles for GRHL2.

Histone deacetylases (HDACs) are enzymes that catalyze the removal of acetyl groups from histones promoting chromatin condensation and loss of gene expression [30]. Of note, HDACs are often aberrantly expressed or dysregulated in GBM [31,32,33,34,35] and are implicated in silencing tumor-suppressor and pro-apoptotic genes [36,37,38]. In GBM, HDAC1 promotes pathogenesis by regulating the PI3K/AKT and MEK/ERK signaling pathways [39]. HDAC enzyme inhibitors (HDACi) have been utilized for the treatment of some cancers [40,41]. In GBM, HDACi increase acetylation of histones and enhance damage induced by DNA alkylating agents such as temozolomide (TMZ) [42]. HDACi can block the cancer cell cycle [43] and alter the cancer stem cell and mesenchymal transcription programs [44,45,46]. HDACi are also capable of promoting the epigenetic activation of genes crucial to apoptosis, cell-cycle arrest, and proliferation [47].

In this study, we tested whether GRHL2 could cooperate with HDACi in GBM to suppress mesenchymal properties and impair cell survival. GRHL2 greatly enhanced the ability of HDACs to repress mesenchymal protein expression and promoted cell death. Unexpectedly, GRHL2 also significantly impacted the cell cycle of mesenchymal GBM cells.

## 2. Materials and Methods

### 2.1. Cell Culture

LN229 (ATCC CRL-2611) and U87-MG (ATCC HTB-14) cells were maintained as an adherent culture in DMEM media (Cytiva, SH30022.02, Washington, DC, USA) supplemented with stable glutamine analog SG-200 (Cytiva, SH30590.01). Media also contained 10% tetracycline-free fetal bovine serum (FBS) (PEAK, PS-FB3), 1% sodium pyruvate (Corning, 25-000-CI, Corning, NY, USA), and 1% penicillin/streptomycin (Cytiva, SV30010) in 10 cm culture dishes. Cell cultures were incubated at 37 °C and 5% CO_2_ in a humidified atmosphere. GRHL2 expression was induced for 72 h using 200 ng/mL of doxycycline (Millipore Sigma, D9891, Burlington, VT, USA). For inducible GSCs, we used 300 ng/mL. After 48 h of dox induction, cells were treated with individual HDAC inhibitors. HDAC inhibitors used were APExBIO’s vorinostat (#A4084-5.1), mocetinostat (#A4089).

GSC cells were obtained from the Mayo Clinic Brain Tumor Patient-Derived Xenograft National Resource [48]. These cells were grown in glioblastoma stem cell media composed of Neurobasal media (Gibco #21103049 New York, NY, USA), with 10 mL B27 without vitamin A (Gibco #12587010). The media also contained MEM non-essential amino acids (Gibco #11140-050), 10 mg EGF (PeproTech AF-100-15 New York, NY, USA), 10 mg FGF (PeproTech 100-18B), 1 mM sodium pyruvate (Gibco #11360-070), 2 mM SG-200 (Cytiva, SH30590.01), as well as 100 units/mL penicillin and 100 mg/mL streptomycin (Gibco #15140122). GSCs were passaged using glioma-stem-cell/brain-tumor-initiating methods. This involves culturing neurospheres in suspension followed by collection via centrifugation at 200× *g* for 5 min for the removal of the media. For passaging, cells were resuspended in 1 mL or less 0.25% trypsin, 2.21 mM EDTA, without sodium bicarbonate (Corning #25-053-Cl) for 3–5 min incubated at RT. Trypsin was then diluted with 10 volumes of complete Neurobasal media, and the cells were centrifuged and washed once more with the Neurobasal medium before returning the cells to the culture.

### 2.2. Generation of Cell Lines via Lentiviral Transduction

Doxycycline-inducible GRHL2 cell lines were created through lentiviral transduction. GRHL2 cDNA was amplified from the SCaBER (ATCC HTB-3) cell line and directionally inserted into a pCW backbone (Addgene, 50661 Cambridge, MA, USA) rtTA-advanced tetracycline-ON vector system using NheI and BamHI sites. The resulting plasmid was sequenced for GRHL2 to ensure error-free cloning. Lentivirus was packaged and amplified in HEK293T (ATCC CRL-3216) (or GSC-media adapted HEK293T [49]) cells by transfection of 0.6 μg of VsVg, 3 μg of δ8.9, and 6 μg of the doxycycline-inducible GRHL2 plasmid and 0.6 μg pRev in a 100 mm^2^ plate. Media containing lentivirus were harvested, filtered through a 0.45 μm filter, mixed with polybrene (1 μg/mL), and then added to GBM cells. After 12 h, cells were given fresh media and then after 48 h cells were trypsinized and then treated initially with 1 mg/mL of puromycin (Cayman Chemical Co., 13884 Ann Arbor, MI, USA). After 48 h, puromycin was reduced to 0.5 μg to accommodate the lower number of surviving cells, before increasing puromycin to 1 mg when cells approached confluency, where it was kept for future cell-line maintenance.

### 2.3. Cell Lysis and Immunoblots

After removing media from cells in a 6-well plate, 1.5 mL of ice-cold PBS was added, and the cells were scraped into a 1.5 mL tube. Cells were collected by centrifugation at 400× *g* for 4 min, PBS was removed, and cells were placed on ice for lysis or stored at −80 °C. Cell pellets were lysed in 10 mM Tris-Cl (pH 8.0), 150 mM sodium chloride, 1% Triton X-100, 1 mM EDTA, and 3% 50× Complete Protease Inhibitor (Sigma #P8340 New York, NY, USA). Samples were incubated on ice for 20 min, then centrifuged for 20 min at 13,000× *g*. Lowry protein estimation was conducted in order to quantify the concentrations of cell lysates with DC™ Protein Assay Reagent S (Bio Rad #500-0115 Hercules, MN, USA) and DC Protein Assay Reagent A (Bio Rad #5000113) according to the manufacturer protocol. Absorbance readings at 750 nm using a BioTek Synergy 2 plate reader were used to calculate mg/mL protein values. Samples were made up of NuPage LDS sample buffer (Invitrogen, NP0007 Carlsbad, CA, USA) and Invitrogen Novex 10× Bolt Sample Reducing Agent (Thermo Fisher Scientific #B0009 New York, NY, USA), incubated at 95 °C for 7 min before loading equal protein amounts onto Mini-PROTEAN TGX Stain-Free Gels 4–20% Tris-Glycine polyacrylamide gels (Bio Rad #4568096). Between 50 and 200 μg of protein was analyzed. Gels were run in TGS buffer for 35 min at 100 volts and transferred to Trans-Blot^®^ Turbo™ Mini Nitrocellulose membranes (Bio Rad #1704158) for 5 min using the Trans-Blot^®^ Turbo™ RTA Transfer Kit, Nitrocellulose System (Bio Rad #170-4270), and Trans-Blot Turbo Transfer Buffer (Bio Rad #10026938). Membranes were blocked in 5% StartingBlock™ (TBS) Blocking Buffer (Fisher #37542) with orbital shaking for 30 min at RT. Primary antibodies were incubated with blots overnight at 4 °C with orbital shaking, followed by antibody removal and washing in 10 mL of TBS-T three times for 5 min at RT with orbital shaking. Primary antibodies were detected by the addition of species-specific horseradish-peroxidase (HRP) conjugated-secondary antibody in 5% nonfat dry milk, incubated with orbital shaking overnight at 4 °C followed by washing as for primary antibodies. After removing the last wash buffer, chemiluminescent visualization (SuperSignal™ West Pico PLUS Chemiluminescent Substrate, Thermo Fisher Scientific, #34578) was used. Membranes were visualized on an Invitrogen iBright 1500 or BioRad ChemiDoc MP. Membranes were then stripped three times in a low-pH stripping buffer (25 mM Glycine, 1% SDS, pH 2.3) for 15 min on an orbital shaker. Membranes were then washed in 10 mL TBS-T for 5 min on the orbital shaker before the new primary antibody was added (Table 1).

### 2.4. Annexin V Flow Cytometry

LN229 cells were seeded 1 × 10^6^ cells in a 10 cm dish. After 56 h of drug treatment, media were removed and placed in a labeled 15 mL falcon tube. Cells were collected through trypsin dissociation, added to the 15 mL tube, and washed twice with PBS without calcium or magnesium. Cell numbers were counted on Countess (Thermo Fisher). Cells were pelleted at 500× *g* for 10 min. The cell pellet was resuspended in Annexin V Binding Buffer (BD Pharmingen, 556454 Franklin USA), and 5 × 10^5^ cells were transferred to a FACS tube containing 5 µL propidium iodide (PI) staining solution (Peprotech 60910-00) and 5 µL FITC-Annexin V (BD Pharmingen, 556419). Cells were incubated for 15 min in the dark. After incubation, 400 µL of Annexin V Binding Buffer was added and FACS was performed on a BD LSRII or LSR Fortessa and analyzed using FlowJo v10. Compensation controls used for this analysis included unstained cells and cells stained for each individual fluorophore (Cy5-PI and FITC-Annexin V). Gating was applied to eliminate cell debris and doublets through forward and side scatter plots.

### 2.5. Cell-Cycle Analysis by Flow Cytometry

GSC and LN229 cells were seeded in 6-well plates and incubated with doxycycline for 48 h, after which doxycycline was refreshed and vorinostat (2.5 μM) was added for 24 h. Following this, the media were removed, and cells were dissociated using trypsin-EDTA. The cells were washed twice with ice-cold 1×PBS. Cell numbers were counted on a Countess (Thermo Fisher) and were pelleted at 200× *g* for 10 min. A total of 2 × 10^6^ cells were resuspended in 1 mL of ice-cold PBS and fixed in 9 mL of 70% ethanol. The cells were stored at −20 °C for 24 h. The cells were then pelleted at 500× *g* for 10 min and washed with 1X PBS. The cells were stained with 400 μL of a staining solution composed of 0.1% *v*/*v* Triton-X 100 in PBS, 2 mg DNAse-free RNAse A (Thermo Fisher EN0531), and 500 μg/mL propidium iodide (Biogems 60910-00 New York, NY, USA) and incubated for 15 min at room temperature. The DNA content was measured on BD LSRII or LSR Fortessa and analyzed using FlowJo v10. Gating on an FSC-A vs. SSC-A plot, FSC-H vs. FSC-W plot, and SSC-H vs. SSC-W plot was used to eliminate debris and any doublets from the data.

### 2.6. Rapid Immunoprecipitation and Mass Spectrometry of Endogenous Proteins (RIME)

After induction and treatment, cells were fixed in 1% formaldehyde in 1× phosphate-buffered saline (PBS) for 10 min. The reaction was quenched with 125 mM for 2–3 min. The cells were washed three times with ice-cold PBS before being scraped in ice-cold PBS and put in a 50 mL conical tube. Cells were spun down at 1000× *g* for 5 min. After decanting the PBS, the pellets were left in the −80 °C freezer overnight. Lysates were prepared via a nuclear extraction, which was performed by adding LB1 (50 mM HEPES-KOH; pH 7.5, 140 mM NaCl, 1 mM EDTA, 10% glycerol, 0.5% NP-40, 0.25% Triton X-100) to the cell pellet, shearing the cellular membrane with a 20-gauge needle, then incubating the lysate at 4 °C for 10 min on the nutator. Lysates were spun down at 2000× *g* for 5 min. The buffer was decanted and the pellet was suspended in LB2 (10 mM Tris-HCl; pH 8.0, 200 mM NaCl, 1 mM EDTA, 0.5 mM EGTA), then rotated for 5 min at 4 °C. Lysates were clarified by centrifugation at 2000× *g* for 5 min. The buffer was decanted, and the pellet (~2 × 10^7^ cells) was suspended in 300 mL of the final lysis buffer, LB3 (10 mM Tris-HCl; pH 8.0, 100 mM NaCl, 1 mM EDTA, 0.5 mM EGTA, 0.1% sodium deoxycholate, 0.5% N-lauroylsarcosine). The nuclear lysates were sonicated in a 1.5 mL Eppendorf tube in a BioRuptor in two 10 min intervals with 30 s “On” and 30 s “Off”. A total of 30 mL of 10% Triton X-100 was added to each sample before clarifying the lysate via centrifugation at 16,000× *g* for 30 min at 4 °C. Protein concentration was determined using the Lowry assay. In total, 20 mg of protein was added to the IP buffer (16.7 mM Tris-HCl, 1.2 mM EDTA, 167 mM NaCl, 1% Triton X-100) (1 mL total per sample) along with the GRHL2 antibody. Samples were rotated overnight at 4 °C. A total of 100 μL of Pierce protein A/G magnetic beads were added to each sample, then rotated for 3 h at 4 °C. Beads were washed 10 times with RIPA2 buffer (50 mM HEPES-KOH; pH 7.6, 1 mM EDTA, 0.7% sodium deoxycholate, 1% NP-40, 0.5 M LiCl), then given to the UAB Mass Spectrometry Core for analysis or the protein was eluted from the beads using elution buffer (100 mM NaHCO_3_, 1% SDS), then subsequently used for confirmation via Western blotting.

Proteins were eluted with DTT and denatured at 70 °C for 10 min prior to being loaded onto a Novex NuPAGE 10% Bis-Tris Protein gel (Invitrogen, NP0315BOX) and separated for 35 min at 200 V. The gels were stained overnight with a Novex Colloidal Blue Staining kit (Invitrogen, LC6025). Following de-staining, the lanes were cut into multiple molecular-weight fractions and equilibrated in 100 mM ammonium bicarbonate. Each gel fraction was then digested overnight with Trypsin Gold, Mass Spectrometry Grade (Promega, V5280 Madison, WI, USA). Peptide extracts were reconstituted in 0.1% formic acid in ddH_2_O at 0.1 mg/mL.

Peptide digests (8 mL) were injected into a 1260 Infinity nHPLC stack (Agilent Technologies, Santa Clara, CA, USA) and separated using a 75-micron I.D. ×15 cm pulled-tip C-18 column (Jupiter C-18 200 Å, 5 micron, Phenomenex). This system runs in-line with a Thermo Q Exactive HFx mass spectrometer, equipped with a Nanospray FlexTM ion source (Thermo Fisher Scientific), and all data were collected in CID mode. The nHPLC was configured with binary mobile phases that include solvent A (0.1% formic acid in ddH2O) and solvent B (0.1% formic acid in 15% ddH2O/85% ACN) and programmed as 10 min at 5% B (2 mL/minute, equilibrate), 90 min at 5%–40% B (linear: 0.5 nL/minute, analyze), 5 min at 70% B (2 mL/min, wash), and 10 min at 0% B (2 mL/minute, equilibrate). Following each parent ion scan (300–1200 *m*/*z* at 60 k resolution), fragmentation data (MS2) were collected on the 10 most intense ions at 7.5 k resolution. For data-dependent scans, charge state screening and dynamic exclusion were enabled with a repeat count of 2, repeat duration of 30 s, and exclusion duration of 90 s.

The XCalibur RAW files were collected in profile mode, centroided, and converted to MxXML using ReAdW v. 3.5.1. The mgf files were then created using MzXML2Search (included in TPP v. 3.5) for all scans. The data were then searched using SEQUEST (Thermo Fisher Scientific), which was set for three maximum missed cleavages, a precursor mass window of 20 ppm, trypsin digestion, variable modification C at 57.0293, and M at 15.9949 as a base setting, with additional post-translational modifications (e.g., Phos, Ox, GlcNAc, etc.) that may be applied at a later time as determined to be of importance experimentally. Searches were performed with a species-specific subset of the UniProtKB database.

### 2.7. Immunofluorescence

Inducible LN229 cells were treated with 200 ng/mL doxycycline every 3 days for 14–21 days. Cells were collected through trypsin-EDTA dissociation and centrifuged at 300× *g* for 5 min to pellet. A total of 10^5^ cells were suspended in 100 µL of 1% BSA in PBS and deposited on slides using CytoSpin, rinsed with PBS, and fixed by incubating with Image-ITTM (Invitrogen) for 15 min at room temperature. The slides were washed 3 times with PBS and then permeabilized with 0.5% NP-40 in PBS for 10 min at room temperature. The slides were rinsed twice with PBS and dehydrated with 1:1 solution of acetone and methanol for 10 min at room temperature. The slide surface was blocked for 30 min using 1% BSA. The cells were incubated overnight with 10 µg/mL anti-CD44 antibody (R&D systems, BBA10) diluted in 1% BSA in PBS at 4 °C. The cells were washed 3 times with PBS and incubated with goat anti-mouse IgG antibody (DyLight 488, GeneTex New York, NY, USA) at a concentration of 1:500 in 1% BSA in PBS for 1 h at 4 °C. The cells were washed 3 times with PBS and dried, and coverslip was mounted using Mounting Medium with DAPI (ibidi). Images were captured with a Nikon Eclipse confocal Ti microscope and NIS Elements imaging software.

## 3. Results

### 3.1. GRHL2 Induction Represses Mesenchymal Protein Expression and Enhances HDAC inhibitor-Mediated Apoptosis

In GBM cells, GRHL2 is typically strongly repressed (Appendix A). These data are consistent with the observation that most GBM cells display an expression of canonical mesenchymal markers such as ZEB1, MMP2, YKL-40, SNAI2, and CD44 (Appendix A). To determine the impact of GRHL2 expression on GBM cells, we used lentiviral transduction to establish doxycycline (DOX)-inducible GRHL2 cells using the GBM LN229 line (LN229^GRHL2^). Upon the addition of doxycycline (LN229^GRHL2+DOX^), these cells displayed inducible GRHL2 protein expression (Figure 1, lanes 1 and 2). We next tested whether the inducible GRHL2 functioned canonically to repress mesenchymal markers. ZEB1 is canonically repressed by GRHL2 in a negative feedback loop in multiple cancer types, including breast cancer [50], bladder cancer [27], and ovarian cancer [51]. We tested protein expression for ZEB1 and the mesenchymal cancer cell proteins MMP2, SNAI2, and CD44 due to their upregulation in GBM vs. normal tissue (Appendix A). Although CD44 did not appear to be affected by GRHL2, both ZEB1 and MMP2 were significantly repressed by the induction of GRHL2 in LN229 GBM cells (Figure 1). We observed a similar response with another cell-culture-adapted GBM line engineered to express inducible GRHL2 (U87^GRHL2^) (Appendix A). These observations confirm the functionality of the engineered inducible GRHL2 expression cassette.

We also assessed GRHL2 function in the presence of an HDAC inhibitor, vorinostat, which inhibits class I, II, and IV HDACs. HDAC inhibitors can reprogram pathogenic mesenchymal transcriptomes. For example, ZEB1 repression via class I HDAC inhibition can resensitize some cancer cells to chemotherapy [52]. We chose vorinostat because it was recently identified in a study of diffuse intrinsic pontine glioma (DIPG) as having a relatively high CNS multiparameter optimization desirability score [53]. This is consistent with prior reports of its moderate blood–brain barrier penetration [54,55,56]. Vorinostat on its own in LN229 cells appeared to lead to a modest increase in expression of some mesenchymal markers consistent with its biochemical function of preventing enzymatic removal of transcriptionally activating histone acetylation marks. However, in conjunction with GRHL2, this effect appeared to be largely abrogated (Figure 1), suggesting that GRHL2 largely dominated over the effects of HDACi in LN229 cells. Observation of cells expressing GRHL2 in the presence of vorinostat did suggest that the combination caused a decrease in cell survival. Given the emerging link between mesenchymal properties and GBM tumor cell survival during therapy (resistance) or following therapy (recurrence) [10,57,58], we tested whether the observed change in mesenchymal proteins with GRHL2 alone was sufficient to trigger detectable levels of apoptosis. We examined common markers of apoptosis, including cleavage of caspase 3 (CC3), phosphorylated H2AX (pH2AX) (resulting from apoptotic DNA fragmentation), and membrane integrity by annexin V binding. We did not observe marked increases in cleaved caspase 3 following GRHL2 induction in LN229 cells (Figure 2A,B,E), indicating that reprogramming GBM cells with GRHL2 is not sufficient to drive appreciable apoptosis. We observed a similar outcome with U87^GRHL2^ cells (Appendix A).

We considered the possibility that a less culture-adapted cell line that closely resembled patient tumors may be more sensitive to epithelial reprogramming by GRHL2. To test this, we constructed GRHL2 dox-inducible glioma stem cells (GSCs) derived from a low-passage PDX sample [48]. These cells are grown under serum-free glioma stem cell culture conditions [59,60], which is considered one of the best current in vitro models available for glioblastoma [44,61]. In GSCs, induction of GRHL2 alone did not lead to marked increases in cleaved caspase 3 (Figure 2D).

Intrigued by GRHL2’s failure to strongly impact cell survival on its own, we wondered whether GRHL2 may sensitize GBM cells to apoptosis under conditions of enhanced transcriptional reprogramming, such as those induced by HDAC inhibitors [44,45,46]. We therefore tested GRHL2 induction in the presence of vorinostat or mocetinostat (a selective HDAC1 and 2 inhibitor).

To monitor this following single or combined treatments, we performed immunoblots for apoptotic markers, including CC3 and pH2AX. Combining GRHL2 expression with HDACi in LN229 cells increased CC3 levels in both LN229 and GSCs (Figure 2A–E). This effect was particularly pronounced in LN229 cells when GRHL2 was combined with mocetinostat. U87^GRHL2^ cells displayed a similar response to the mocetinostat combination treatments (Appendix A).

### 3.2. GRHL2 Expression in GBM Causes Cell-Cycle Defects

Because HDACs are well-established modulators of the cell cycle, we tested whether GRHL2 altered these effects. We first examined whether GRHL2 had any impact on the expression of cell-cycle regulatory proteins such as pRB, cyclin B1, pCDC2, and p21 in LN229 cells. Unexpectedly, GRHL2 on its own, and also in combination with HDACi, appeared to impact expression of these markers, particularly cyclin B1 (Figure 3A,B). Changes observed in p21 and pCDC2 suggest that GRHL2 may also affect these cell-cycle markers, although in our study these did not achieve statistical significance (Figure 3A,B). To determine the impact of GRHL2 alone on cell-cycle progression, we examined LN229 for DNA content using PI staining and flow cytometry. These studies revealed an increase in S-phase cells compared to controls (Figure 3C).

Inappropriate or unscheduled entry into the S phase can create replication stress, suggesting that long-term expression of GRHL2 in these cells may lead to gross cell-cycle defects. Therefore, to assess if GHRL2 might have such consequences, we treated parental inducible LN229 cells with doxycycline for 14 days. After two weeks with doxycycline, we observed the appearance of a population of cells that appeared to have greater than 4N (Figure 3D). To visualize multinucleated cells formed as a result of GRHL2 induction, we treated cells with doxycycline for three weeks followed by staining cell membranes for CD44 and for nuclei with DAPI. These experiments revealed an increase in multinucleated cells in the GRHL2-induced cells (Figure 3E).

We next examined GSCs for the effect of GRHL2 on the cell-cycle protein expression and DNA content by PI/flow cytometry. In GSCs, induction of GRHL2 significantly impacted cell-cycle protein expression when combined with HDAC inhibition (Figure 4A), simultaneously promoting the signals for cell-cycle progression (cyclin B1) and cessation (p21). This dysregulated cell-cycle signaling was associated with changes in DNA content after just 3 days of GRHL2 induction (Figure 4B). This effect was not apparent when parental GSCs were treated in the same way to control for the effect of doxycycline (Appendix A), indicating that the effect was a result of GRHL2 expression.

Together, these observations suggest that introducing this epithelial transcription factor impacted the ability of these cells to correctly undergo cell division. Given that GRHL2 has roles in establishing apical–basal polarity [22]—which is typically absent in mesenchymal cells—as well as roles in spindle pole orientation, we hypothesized that GRHL2 might physically interact with the mitotic machinery. To test this, we performed rapid immunoprecipitation of GRHL2 followed by mass spectrometry [62] in GRHL2-inducible GBM cells with vs. without doxycycline treatment. These analyses revealed that in immunoprecipitated material, GRHL2 induction increased the presence of several proteins with key mitotic roles (Appendix A), suggesting that GRHL2 may physically interact with these proteins (Appendix A).

Given the known roles of GRHL2 in mitosis, and our observations that expression in GBM cells alters normal cell division, we hypothesized that a pan-cancer analysis of cells might indicate that GRHL2 expression associates with higher levels of aneuploidy in tumors. We therefore analyzed patient tumor GRHL2 expression data from The Cancer Genome Atlas with respect to the reported aneuploidy score for those tumors (Figure 4C). We observed that around an expression level of 6 RSEM, there appeared to be a bimodal distribution of GRHL2 expression, possibly reflecting the tumor types that are either epithelial (GRHL2 high) or mesenchymal (GRHL2 low). Partitioning the data at this median value revealed a modest yet significant pan-cancer relationship between high GRHL2 expression and increased levels of aneuploidy (Figure 4D). Thus, our observations of GRHL2 impacting the cell cycle, together with GRHL2’s known role in mitosis, suggest that aberrant GRHL2 expression has the potential to contribute to cancer cell aneuploidy.

## 4. Discussion

In this study, we capitalized on the fact that GRHL2 is not detectably expressed in GBM to test whether this epithelial transcription factor can cooperate with HDACi to revert the mesenchymal phenotype and impair GBM survival. GRHL2 represses mesenchymal factors in a variety of cancers, including breast [50,63], ovarian [26] colorectal [28,64], and gastric cancers [20]. We found that GRHL2 induction in GBM is also capable of disrupting the mesenchymal properties. In addition, GRHL2 promoted the apoptotic effects of HDAC inhibitors, which is consistent with previous reports that reverting the mesenchymal state of GBM may promote therapeutic sensitivity [36,45].

A surprising result from our study was the impact of GRHL2 on cell-cycle progression and genome copy number. There are several possible contributors to GRHL2-mediated disruption of normal cell division in GBM. Where GRHL2 is normally expressed in the epithelium, GRHL2 functions in the establishment of polarity [22], which is important for the positioning of the mitotic spindle [23]. In proliferating epithelia, cells arrange their mitotic spindle in the plane of the epithelium [65]. Mesenchymal cells, on the other hand, display very different polarity dynamics than epithelial cells [66]. The exact mechanisms controlling the positioning of mitotic spindles in mesenchymal cells is not well understood, but given the role of GRHL2 in establishing cellular polarity, exogenous expression could disrupt such mechanisms. Our mass spectrometry data of GRHL2 immunoprecipitation indicated the presence of critical mitotic regulators, suggesting that introducing GRHL2 into mesenchymal cells may disrupt the spatial arrangements of mitotic machinery, causing defects in cytokinesis and/or abscission, in turn leading to aneuploidy. In support of this, similar to our results in LN229 cells, GRHL2 was shown to accelerate cell-cycle progression of colorectal cancer cells, leading to an aberrant increase in cells in the S and G2/M phases [28].

From a transcriptional perspective, GRHL2 can also upregulate the mitotic cell-cycle regulator MCIDAS [22]. MCIDAS interacts with geminin, which is required for limiting DNA replication to once per cell cycle. Thus, GRHL2 may also transcriptionally alter mesenchymal cells in ways that interfere with appropriate mitotic progression.

One caveat to our study is that the GSCs were generated by lentiviral transduction, which includes a two-week puromycin selection period to obtain a population of cells expressing inducible GRHL2. Although maintained under stem growth conditions, some changes to the cells may have taken place, which may limit the generalizability of our findings to other GSCs. This limitation is inherent to this model, as GSCs are poorly transfectable in transient experiments. Future studies aimed at characterizing the resulting GSCs (marker expression, pathogenicity) would potentially be informative regarding the effects of this process.

Our findings suggest that exogenous GRHL2 in mesenchymal cells adds to cell-cycle stresses induced by HDACi. Such cumulative effects of GRHL2 and HDACi on the cell cycle may enhance cellular sensitivity to apoptosis. We observed that combining HDACi with GRHL2 expression caused p21 upregulation, which promotes cell-cycle arrest; however, GRHL2 induction also causes increases in cyclin B1, which is a counterposing signal to advance the cell cycle. Such conflicting signals may have the potential to sensitize cells to apoptosis.

## 5. Conclusions

The introduction of GRHL2 into GBM cells likely promotes partial mesenchymal-to-epithelial (MET) transition and renders these cells sensitive to apoptosis induced by HDACi, potentially in part though disrupting mitosis (Figure 5). This work provides novel mechanistic insights into reprogramming the pathogenic features of mesenchymal cells by targeting transcription with both small molecules and genetic tools.

## Figures and Tables

**Figure 1 genes-14-01787-f001:**
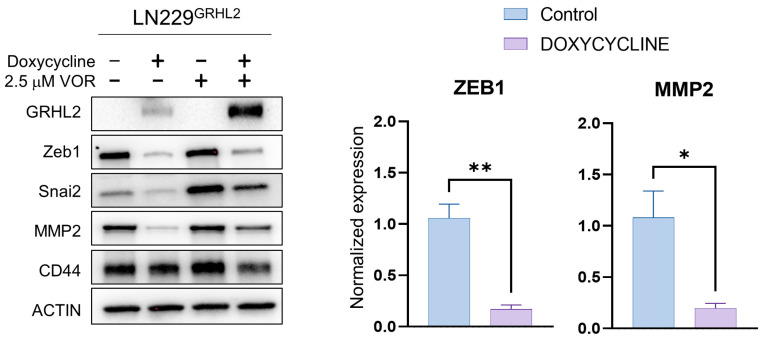
GRHL2 represses mesenchymal proteins in GBM cells. Western blots for GRHL2, Slug, CD44, MMP2, and ZEB1 in LN229 cells with or without doxycycline (200 ng/mL) or 2.5 µM vorinostat (VOR). β-Actin was used as a loading control for protein expression. Graphs depict significant changes in protein expression levels from A. Graphs depict means +/− SEM for *n* = 3. * *p* < 0.05 *t*-test; ***p* < 0.01 *t*-test.

**Figure 2 genes-14-01787-f002:**
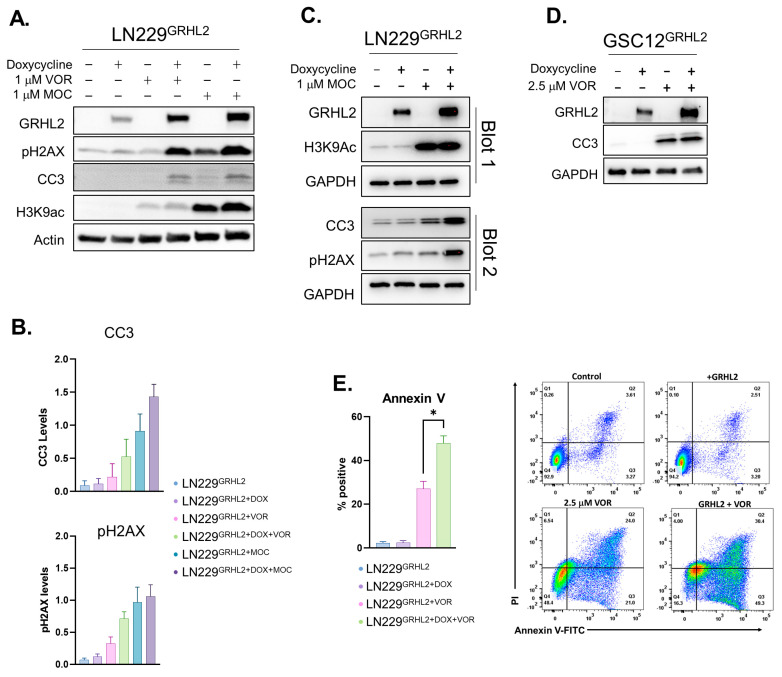
GRHL2 enhances apoptosis of HDACi-treated GBM cells. (**A**) Western blots for CC3, GRHL2, histone 3 lysine 9 acetylation (H3k9ac), and pH2AX in inducible LN229 cells treated with the individual HDACi vorinostat (1 μM) or mocetinostat (1 μM). β-Actin was used as a loading control for protein expression. (**B**) Quantification of LN229’s CC3 and pH2AX in A. (**C**) GRHL2 induction in LN229 cells after 72 h of doxycycline induction, treated for 24 h with mocetinostat (1 μM) followed by immunoblot analysis of CC3, pH2AX, and acetylation of histone 3 lysine 9. Data from blots 1 and 2 are the same samples on different membranes. (**D**) CC3 levels in GRHL2-inducible glioma stem cells from GBM PDX after doxycycline with or without vorinostat (2.5 μM). (**E**) Apoptosis as measured by annexin V in doxycycline-induced LN229 cells treated with 2.5 μM vorinostat for 56 h. Graphs depict means + SEM for *n* = 3. * *p* < 0.05 *t*-test.

**Figure 3 genes-14-01787-f003:**
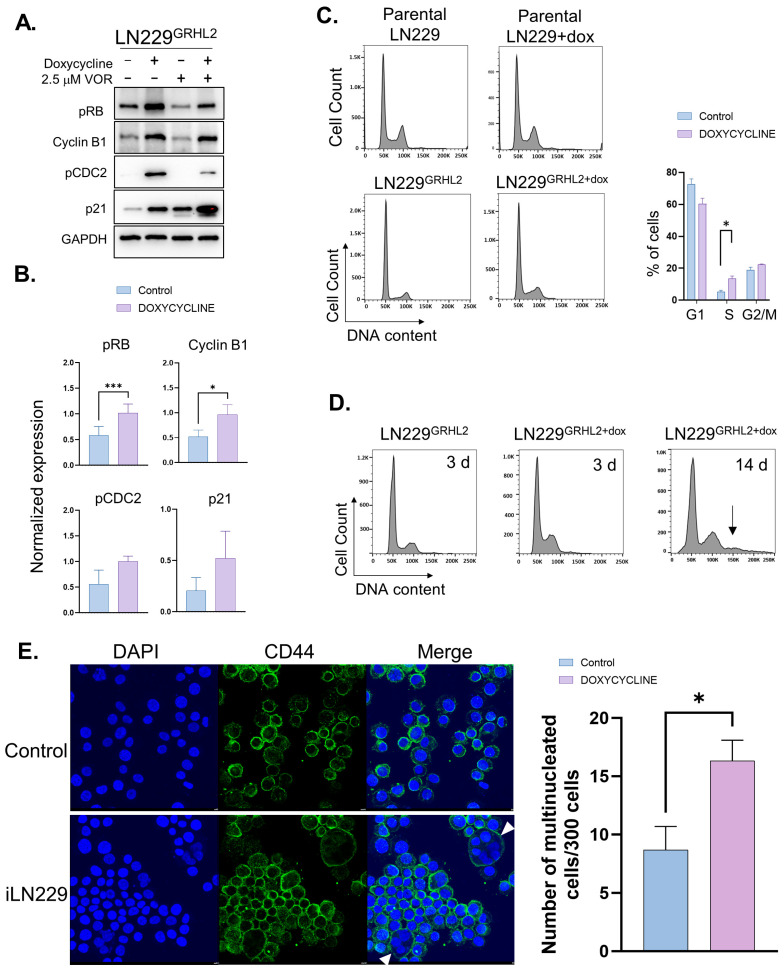
Exogenous GRHL2 expression alters the GBM cell cycle. (**A**) Inducible LN229 cells were treated with dox for 72 h with or without HDACi for the last 24 h and were immunoblotted for indicators of cell-cycle phases pCDC2, pRB, cyclin B1, and p21. GAPDH was used as a loading control for protein expression. (**B**) Quantification of data in A. (**C**) Cell-cycle (propidium iodide) flow cytometry of parental or inducible LN229 treated with dox for 72 h. Graph displays mean values for G1, S, and G2/M phase. (**D**) Cell-cycle (propidium iodide) flow cytometry of iLN229 treated with dox for 3 or 14 days. Arrow indicates > 4 N cells. (**E**) Control or dox-treated (21 days) iLN229 cells were stained with CD44 to visualize the cell membrane and DAPI to visualize nuclei. Cells were then counted for multinucleated cells. White arrowheads identify multinucleated cells. Mean values of multinucleated cells are per 300 cells counted. Graphs depict means +/− SEM for *n* = 3. * *p* < 0.05 *t*-test; ** *p* < 0.01 *t*-test; *** *p* < 0.001 *t*-test.

**Figure 4 genes-14-01787-f004:**
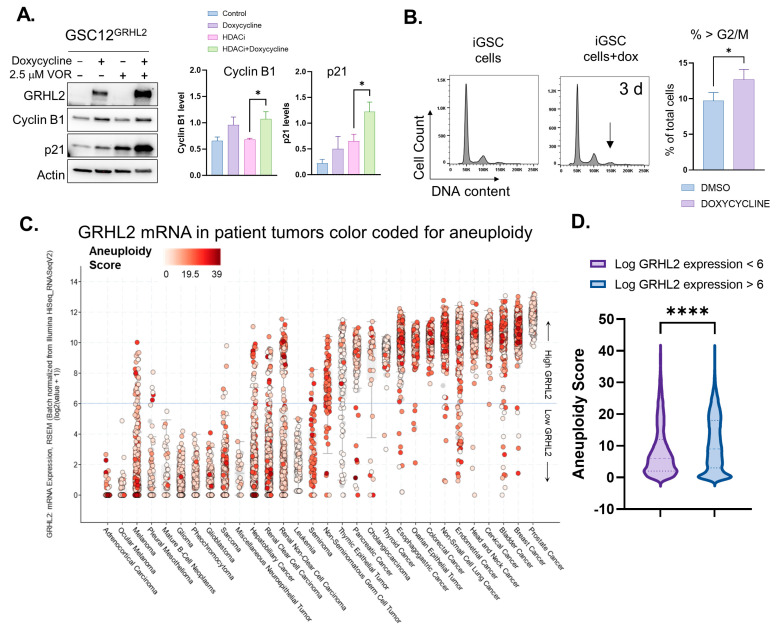
GRHL2 expression dysregulates cell cycle in glioma stem cells and associates with aneuploidy in patient tumors. (**A**) Inducible glioma stem cells (GSCs) were treated with dox for 72 h with or without HDACi for the last 24 and were immunoblotted for indicators of cell-cycle phases cyclin B1, and p21. β-Actin was used as a loading control for protein expression. (**B**) Cell-cycle (propidium iodide) flow cytometry of inducible GSCs treated with dox for 72 h. * *p* < 0.05, *t*-test. Arrow indicates > 4 N cells. (**C**) GRHL2 mRNA expression in patient tumors by cancer type, arranged by mean expression level and color-coded for aneuploidy score. (**D**) GRHL2 mRNA expression in cancers vs. aneuploidy score. (*n* = 3077 scores < 6, *n* = 6579 > 6.) Data in C and D were captured from The Cancer Genome Atlas (TCGA) cBioBortal. **** *p* < 0.0001 *t*-test.

**Figure 5 genes-14-01787-f005:**
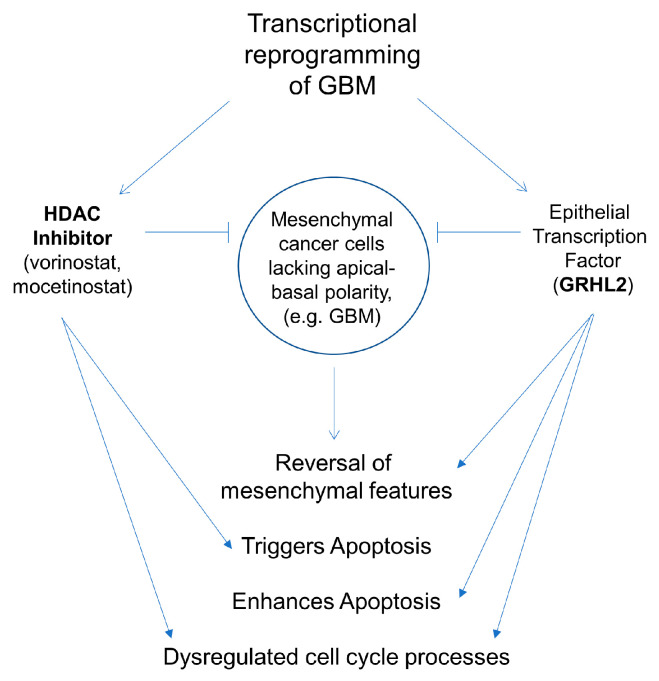
GRHL2 expression in glioblastoma cells. GRHL2 suppresses mesenchymal factor expression. HDACi dysregulation of the cell cycle is further promoted by GRHL2, increasing the chance that cells will undergo apoptosis.

**Table 1 genes-14-01787-t001:** Antibodies.

Antibody	Catalogue Number
BIM	Cell Signaling Technologies, 2933
CD44	R&D Systems, BBA10
Cleaved Caspase 3	Cell Signaling Technologies, 9664
GRHL2	Invitrogen, PA5-28973
SNAI2	Cell Signaling Technologies, 9585
H3K9ac	Cell Signaling Technologies, 9766
MMP2	Cell Signaling Technologies, 40994
ZEB1	Cell Signaling Technologies, 70512
b-Actin	Santa Cruz, sc-47778
pH2AX	Cell Signaling Technologies, 9718
pRB	Cell Signaling Technologies, 9308
Cyclin B1	Cell Signaling Technologies, 12231
TK1	Cell Signaling Technologies, 28755
pCDC2	Cell Signaling Technologies, 4539
P21 Cell	Signaling Technologies, 2947
Cyclin E1	Cell Signaling Technology, 20808
CD44	R&D Systems, BBA10

## Data Availability

Data are contained within the article or Appendix A.

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
