# Peer review of "Enhancing Transcriptional Reprogramming of Mesenchymal Glioblastoma with Grainyhead-like 2 and HDAC Inhibitors Leads to Apoptosis and Cell-Cycle Dysregulation"

_genes, 2023, doi:10.3390/genes14091787_

Round 1

Reviewer 1 Report

Reviewer has following minor comments for the authors:

Lines 88-98: Sentence needs to be fragmented to enhance readability.

Line 124: EDTA and 3% 50X Complete Protease Inhibitor (Sigma #P8340. Samples were incubated  à bracket needs to be closed

Line 148: grammatical error

Line 152: table should be numbered. All text should be aligned.

Line 170 – 171: grammatical error

Figure 1. The authors have not commented on CD44 protein expression not being related to induced expression of GRHL2.

Figure 1. While caption mentions A., neither figure nor text indicate Figure 1A or 1B.

Figure 1. Why is expression of canonically repressed proteins upon GRHL2 induction higher upon administration of Dox with VOR?

Figure 2. Correct the legend à Mean plus minus SEM.

Fig2 Why was annexin V not measured for PDAX

Figure 4. What about the expression of pCDC2 and pRB in GSC upon induction of GRHL2 with and without HDACi?

Figure 4b. %S and %M not reported.

Discussion: The discussion at present is merely a summary of the findings. The significance and novelty of these findings with reference to published literature should be included. The discussion can be expanded to include the limitations of the study. Future directions can be discussed as well. 

Reviewer 2 Report

Minor comments:

1) Line 32 is DNA alkylating agent temozolomide??. Only substance used in chemotherapy. If not. Clarify?. Name all with reference.

2) Line 37, name major critical drivers of mesenchymal phenotypes responsible for heterogeneity in GBM tumors. If possible, have a table of these drivers ?.

3) Line 43, 47, 50, 61. reference missing?.

4) Line 270, Reason for choosing these markers protein for investigation.

Other genes Snail1 and Slug, Twist1 and Twist2, MMps are not studied

5) N Cadherin levels are also not checked.

6) Write the molecular weight of the protein on all the genes.

7) Please show pro and cleaved caspase 3 on the western blot.

8) Figure 3B pCDC2 and P21 are not significant??

9) Figure 3E provides a wide-field image for the image to observe the actual phenotype.

10) Please improve discussion section by adding details. There are instance when the sentence are long and are not related to the next sentence.

11) In material and Methods, please remove extra details.

Good

Reviewer 3 Report

In the manuscript by Kotian et al., the authors assess the role of GRHL2 in the EMT phenotype of GBM cell lines. Overall, the authors find that the combination of GRHL2 expression with HDAC inhibitors can increase cell cycle dysregulation, aneuploidy, and apoptosis.

Comments:

Many experiments lack the control cell line treated with doxycycline or its shown only in the supplement. This is an important control that should be included to ensure that the observed effects are due to GRHL2 expression and not dox particularly in combination with HDAC inhibitors. The data, if already present, should be moved from the supplement to the main manuscript.

Figure 1, an description for the rationale behind using vorinostat in this experiment should be included in the text along with the description of the results.

Why was vorinostat chosen over other HDAC inhibitors? Were other standard of care drugs tested such as temozolomide?

Fig 3D, the percentage of cells >4N should be shown, along with control cells treated with dox.

Fig 3e, a larger image with improved resolution should be included.

Fig 4, does the inducible GSC cell line have any changes in stem cell characteristics as compared to the parental?

Fig S5 should be moved to the main manuscript.

Fig 4D, how was the expression cutoff of 6 chosen? Details should be included.

Round 2

Reviewer 2 Report

The reviewer has not replied to the comments successfully.

Minor comments: 

1) Line 37, name major critical drivers of mesenchymal phenotypes responsible for heterogeneity in GBM tumors. If possible, have a table of these drivers. It isn't easy to follow the results. Please have a table with indicators like -,--,--- or +,++,+++ with protein names and functions.

2) Figure 3B pCDC2 and P21 are not significant??.. Please mention this in the text. This trend is suggestive only.

3) Figure 3E provides a wide-field image for the image to observe the actual phenotype. Quantification needed.

Reviewer 3 Report

The authors have addressed my concerns and the manuscript has been improved.

Author Response

We would like to thank the reviewer for their time and thoughtful comments on our manuscript.